# The Role of ER Stress in Diabetes: Exploring Pathological Mechanisms Using Wolfram Syndrome

**DOI:** 10.3390/ijms24010230

**Published:** 2022-12-23

**Authors:** Shuntaro Morikawa, Fumihiko Urano

**Affiliations:** 1Department of Medicine, Division of Endocrinology, Metabolism, and Lipid Research, Washington University School of Medicine, St. Louis, MO 63110, USA; 2Department of Pediatrics, Graduate School of Medicine, Hokkaido University, Sapporo 060-8638, Japan; 3Department of Pathology and Immunology, Washington University School of Medicine, St. Louis, MO 63110, USA

**Keywords:** ER stress, endoplasmic reticulum, β-cell dysfunction, Wolfram syndrome, WFS1, type 2 diabetes

## Abstract

The endoplasmic reticulum (ER) is a cytosolic organelle that plays an essential role in the folding and processing of new secretory proteins, including insulin. The pathogenesis of diabetes, a group of metabolic disorders caused by dysfunctional insulin secretion (Type 1 diabetes, T1DM) or insulin sensitivity (Type 2 diabetes, T2DM), is known to involve the excess accumulation of “poorly folded proteins”, namely, the induction of pathogenic ER stress in pancreatic β-cells. ER stress is known to contribute to the dysfunction of the insulin-producing pancreatic β-cells. T1DM and T2DM are multifactorial diseases, especially T2DM; both environmental and genetic factors are involved in their pathogenesis, making it difficult to create experimental disease models. In recent years, however, the development of induced pluripotent stem cells (iPSCs) and other regenerative technologies has greatly expanded research capabilities, leading to the development of new candidate therapies. In this review, we will discuss the mechanism by which dysregulated ER stress responses contribute to T2DM pathogenesis. Moreover, we describe new treatment methods targeting protein folding and ER stress pathways with a particular focus on pivotal studies of Wolfram syndrome, a monogenic form of syndromic diabetes caused by pathogenic variants in the *WFS1* gene, which also leads to ER dysfunction.

## 1. Introduction

The endoplasmic reticulum (ER) is an intracellular organelle that is known for its role in protein folding, calcium storage, and lipid metabolism. The ER lumen is uniquely equipped to ensure the proper folding and maturation of newly synthesized secretory and transmembrane proteins. Pathological insults that perturb ER homeostasis, such as high protein demand, viral infections, environmental toxins, inflammatory cytokines, decreased ER calcium levels, and mutant protein expression, lead to an accumulation of misfolded and unfolded proteins in the ER lumen, a condition termed ER stress [1]. Cells adaptively counteract ER stress by triggering the unfolded protein response (UPR). The UPR has three major regulators: activating transcription factor 6α (ATF6α), protein kinase R-like ER kinase (PERK), and inositol requiring enzyme 1α (IRE1α) (Figure 1) [2,3]. IRE1α forms oligomers on sensing ER stress, followed by autophosphorylation and acquisition of RNase activity [4]. The activated IRE1α splices the *XBP1* intron and induces the expression of the transcription factor, spliced *XBP1* (s*XBP1*) [5,6,7]. Consequently, ER proteins are translocated, folded, and secreted. IRE1α also suppresses protein folding loads by degrading mRNA. This is called regulated IRE1-dependent decay (RIDD) [8]. In addition, IRE1α activates the c-Jun N-terminal kinase (JNK) pathway by binding to TRAF2, which in turn activates the autophagy pathway [9]. PERK can transiently arrest protein synthesis by phosphorylating eukaryotic translation initiation factor 2 subunit α (eIF2α). Phosphorylated eIF2α also induces expression of the transcription factor ATF4 [10], which regulates redox homeostasis, amino acid metabolism, protein synthesis, apoptosis, and autophagy. ATF6α (p90) is transported from the ER to the Golgi apparatus upon sensing ER stress and cleaved [11]. Cleaved ATF6α (p50) translocates to the nucleus and acts as a transcription factor, regulating the transcription of ER chaperones.

The UPR acts as a binary switch. Under physiological conditions, UPR-regulated survival mechanisms outweigh death mechanisms, and proper protein folding in the ER is achieved, including the folding of proinsulin in pancreatic β-cells. Under pathological conditions, the death factors regulated by the UPR pathway such as C/EBP-homologous protein (CHOP) outplay its survival pathways, leading to cell death. This process is called pathological or terminal UPR [16], and sterile inflammation induced by thioredoxin-interacting protein (TXNIP) plays a critical role in this process [17,18]. Given the many vital and complex functions of the ER, ER failure can trigger a range of diseases. Dysregulation of ER homeostasis underlies pancreatic β-cell dysfunction in type 1 diabetes (T1DM) and type 2 diabetes (T2DM) as well as in monogenic forms of diabetes, including Wolfram syndrome.

In this review, we discuss (1) the relationship between ER stress and pancreatic β-cell dysfunction in diabetes, (2) UPR-related gene variants as risk factors for the development of T2DM, (3) the role of the *WFS1* gene and its contribution to the development of Wolfram syndrome and T2DM, and (4) the opportunities for clinical application of ER stress reducers and induced pluripotent stem cell (iPSC)-based treatment against T2DM and monogenic forms of diabetes.

## 2. ER Stress in Diabetes and Pancreatic β-Cell Dysfunction

T1DM is a chronic autoimmune-mediated disease in which pancreatic β-cells are targeted [19,20]. Inflammation localized to pancreatic islets (insulitis) followed by pancreatic β-cell disruption, makes β-cells unable to produce insulin [19], which is necessary to normalize blood sugar levels. In pre-diabetic non-obese diabetic (NOD) mice, an accepted mouse model of T1DM, ER stress markers are augmented in islets [21], suggesting a close relationship between ER stress and the development of T1DM. In addition, immune cells recognize post-translational modifications on β-cell proteins, which are added by enzymes activated under conditions of high cytosolic Ca^2+^ concentrations and ER stress [22]. Furthermore, in a virus-induced diabetes rat model, IRE1α, one of the main regulators of the UPR pathway, is activated before the initiation of insulitis [23]. Knocking out IRE1α in NOD mice leads to pancreatic β-cell dedifferentiation prior to insulitis, and these dedifferentiated pancreatic β-cells evade autoimmune destruction [24]. As supporting evidence, the expression level of CHOP, the molecule that mediates ER stress-induced apoptosis, is increased in the islets of T1DM patients [25], and the proinsulin to C-peptide ratio, used as a biomarker of ER stress, is elevated in new-onset T1DM children [26]. Altogether, these data point to ER stress as an essential and early driver of autoimmune-mediated T1DM in both rodent models and human patients.

T2DM is characterized by insulin resistance and hyperinsulinemia followed by pancreatic β-cell dysfunction and death. Obesity is a major risk factor for the development of T2DM. Dietary- or genetic-induced obesity in mice is known to cause ER stress, which induces IRE1α-dependent activation of JNK. Activated JNK phosphorylates serine 307 of insulin receptor substrate 1 (IRS-1), a substrate for the insulin receptor tyrosine kinase. In turn, the phosphorylation level of Akt, the distal cascade in insulin signaling, is suppressed, and peripheral insulin resistance develops [27]. Insulin synthesis is thought to be enhanced in pancreatic β-cells in the early stages of T2DM to compensate for insulin resistance. Consequently, proinsulin misfolding occurs also in the early stage of T2DM pancreatic β-cells [28]. It has been reported that serum ER stress markers are high in patients with T2DM [29]. Pancreatic β-cell dysfunction is caused by different mechanisms in T1DM, which is attributable to autoimmunity, and in T2DM, which is known to be induced by insulin resistance. However, the excess ER stress in pancreatic β-cell or peripheral tissue is involved in the pathogenesis of both T1DM and T2DM.

## 3. Genetic Risk Factors for T2DM: Findings from GWAS and the Involvement of UPR Genes

T2DM is a multifactorial disease caused by both genetic and environmental factors. Therefore, it is difficult to identify the risk of developing T2DM based only on genetic factors. However, examining genetic risk factors and the molecular mechanisms by which such mutations induce pathological changes that contribute to the development of T2DM aids in our understanding of the heterogeneous nature and pathogenesis of T2DM. Genome-wide association studies (GWAS) provide a method to search for genetic predispositions for diseases using genome-wide single nucleotide polymorphism (SNP) arrays. Large-scale GWAS approaches gathered data from multiple populations and identified more than 500 SNPs associated with T2DM development [30,31]. Interestingly, SNPs occurred in genes that transcribe proteins related to the UPR. In this section, we will first discuss how the dysfunction of molecules that compose the UPR is involved in the pathogenesis of diabetes. Next, we will discuss the association between T2DM and SNPs in UPR-related genes identified by GWAS.

### 3.1. Monogenic and Syndromic Diabetes Derived from UPR Impairment

The variants identified in UPR-related genes known to cause monogenic diabetes are listed in Table 1. Walcott–Rallison syndrome (WRS, OMIM: 226980) is a relatively well-known disease associated with dysfunction of the *PERK* (*EIF2AK3*) gene [32,33,34]. In addition to the development of diabetes in early infancy, WRS has a variety of manifestations, such as skeletal phenotypes (multiple epiphyseal dysplasia, multiple epiphyseal-metaphyseal dysplasia), hepatic dysfunction, and impaired renal function [33,35]. There are no reported cases of diabetes due to pathogenic variants in the *IRE1* (*ERN1*) gene; however, *IRE1α* knockdown in C57BL/6 mice induces diabetes via a mechanism that is independent of autoimmune-mediated β-cell death [36,37]. Mesencephalic astrocyte-derived neurotrophic factor (*MANF*) encodes a secretory neurotrophic factor, originally discovered in astrocytes, that localizes to the ER and indirectly regulates the UPR. ER calcium depletion and ER stress increase *MANF* expression and secretion [38,39,40,41]. ER stress is elevated in the pancreatic β cells of *Manf*-knockout mice, resulting in ER stress-induced cell death and diabetes [42]. Recently, childhood-onset diabetes and a neurodevelopmental delay caused by the *MANF* gene variant have been reported [43]. Those gene disruptions strongly indicate a relationship between abnormalities in the UPR and dysfunction of pancreatic β-cells. There is no clear explanation for the differences in symptoms caused by functional mutations in UPR-related genes. Differences in the expression levels between tissues or the role of each molecule in pancreatic β-cells are speculated to influence the development of diabetes.

### 3.2. SNPs in UPR-Related Genes and Their Contribution to the Development of T2DM

SNPs in UPR-related genes are thought to increase the risk for T2DM development. The GWAS studies have identified SNPs in *ATF6*, the gene with no previously reported diabetes-associated pathogenic variants [58,59] (Figure 1 ①). For example, a study in a population of Pima Indians reported that SNPs in *ATF6* affect T2DM development and serum insulin levels in the oral glucose tolerance test (OGTT) [12]. Studies in Dutch Caucasian populations also reported that *ATF6* SNPs are significantly associated with glucose intolerance and the development of T2DM [13]. On the other hand, studies in Caucasian, African American, and Han Chinese populations have shown that SNPs in *ATF6* did not significantly impact the development of T2DM [60,61]. There are studies on Han Chinese populations indicating that *PERK* SNPs are correlated with an increased risk of prediabetes, elevated BMI, and insulin resistance [14] (Figure 1 ②). A recent GWAS focused on Han Chinese populations reported a high frequency of *XBP1* SNPs in the T2DM patient group and impaired glucose regulation indicated by the OGTT [14,15] (Figure 1 ③). However, there are no other reports showing that SNPs in *PERK*-related genes affect the development of T2DM. The association between UPR dysfunction and the development of diabetes is evident; however, it remains unclear why the risk of developing T2DM differs by SNPs in UPR-related genes and race.

### 3.3. Wolfram Syndrome and WFS1-Related Disorders: A Prototype of ER-Related Disease

We have focused on the role of *ATF6*, *PERK*, and *IRE1α* as UPR-related genes responsible for monogenic diabetes or disease susceptibility in T2DM. In this section, we will describe the genetic basis for Wolfram syndrome, a well-recognized prototype of ER stress-related disease, and the relationship between its causative gene, *WFS1* and T2DM risk.

Wolfram syndrome (OMIM: 222300) is a rare autosomal recessive disorder characterized by insulin-dependent diabetes, optic nerve atrophy, diabetes insipidus, hearing loss, and neurodegeneration [62,63]. Diabetes in Wolfram syndrome presents in patients around six years of age, and optic nerve atrophy manifests around age ten. Diabetes insipidus, neurogenic bladder, obstructive sleep apnea, and deafness may also develop in the next two decades of life, along with brainstem symptoms and cerebellar atrophy, such as dysphagia, ataxia, and central sleep apnea [62,64,65,66,67,68].

Wolfram syndrome is caused by recessive pathogenic variants in the Wolfram syndrome 1 (*WFS1*) gene [49]. Approximately 200 *WFS1* pathogenic variants have been identified in Wolfram syndrome. The *WFS1* gene encodes WFS1 (also known as wolframin), a protein localized to the ER membrane that interacts with ATF6 to regulate the UPR during ER stress [69,70]. WFS1 also plays a critical role in maintaining ER and cytosolic Ca^2+^ homeostasis [1,71], regulating sterile inflammation [72], and insulin granule acidification in pancreatic β-cells [73]. Recently, it has been shown that WFS1 functions as a vesicular cargo protein receptor for exporting secretory proteins from ER [74].

Wolfram syndrome has been recognized as a spectrum disorder since recessive *WFS1* variants can vary in clinical severity [75]. In contrast, some dominant de novo *WFS1* variants are known to cause severe neonatal-onset diabetes, congenital sensorineural hearing loss, congenital cataracts, and developmental delays, which are known as *WFS1*-related disorders [75,76]. There are also known *WFS1*-related disorders characterized by developing only one or few symptoms seen in Wolfram syndrome. For example, a dominant *WFS1* p.Trp314Arg variant causes mild diabetes in an autosomal dominant fashion [77]. In all cases, however, pathogenic *WFS1* variants cause pathological ER stress in pancreatic β-cells, neurons, retinal ganglion cells, and oligodendrocytes, leading to dysfunction and degeneration of affected tissues [69,70,78,79,80]. Wolfram syndrome and *WFS1*-related disorder, which result from pathogenic ER stress caused by a single gene abnormality, have been well-recognized as a prototype of human ER-related disease [70,81].

*WFS1* variants are typically associated with and discussed as the drivers of Wolfram syndrome; however, *WFS1* variants have additionally been identified as risk factors for the development of T2DM. Variants with a reported increased risk for T2DM include rs10010131, rs734312, rs6446482, and the c.1672C > T, p.Arg558Cys missense variant [82,83,84]. The *WFS1* p.Arg558Cys missense variant is associated with mild syndromic manifestations of Wolfram syndrome. Our study in collaboration with Dor Yeshorim (https://doryeshorim.org/, accessed on 1 April 2022) revealed a high *WFS1* p.Arg558Cys allele frequency (2.32%) in the Ashkenazi Jewish population [85]. Although *WFS1* is the responsible gene for the rare Wolfram syndrome, some of its variants may contribute to the common T2DM development.

## 4. Therapeutics Targeting ER Stress as Treatments for Wolfram Syndrome and T2DM

*WFS1* is a locus of broad interest to Wolfram syndrome or Wolfram syndrome-related disorders, including T2DM. Developing a treatment for Wolfram syndrome may provide clues to finding a new therapeutic approach for T2DM. In this section, we describe treatments for Wolfram syndrome, including pharmacological agents targeting the ER and protein folding components as well as regenerative therapies.

### 4.1. Pharmacological Agents to Treat Wolfram Syndrome and T2DM

The link between UPR-related genes and increased risk for Wolfram syndrome and T2DM advocates for the use of ER stress-targeting drugs as therapies for these metabolic disorders.

Dantrolene sodium, a US Food and Drug Administration (FDA)-approved drug for malignant hyperthermia and muscle spasm, stabilizes ER calcium by targeting ryanodine receptors. Dantrolene sodium treatment can prevent cell death in neural progenitor cells differentiated from Wolfram syndrome patients induced pluripotent stem cell (iPSC) [80]. We recently conducted the dantrolene sodium phase 1b/2 clinical study on Wolfram syndrome patients (National Clinical Trial (NCT) number, NCT02829268). Pancreatic β-cell function improved in a select few Wolfram syndrome patients, and there was a significant correlation between baseline β-cell function and change in β-cell responsiveness [86]. Although its effects were limited, dantrolene sodium was the first therapeutic candidate compound for Wolfram syndrome found by high-throughput screening. Intracellular Ca^2+^ stabilization is also a central feature of another Wolfram syndrome therapeutic, the phosphodiesterase 4 (PDE4) inhibitor ibudilast. Ibudilast rescues cell viability and cytosolic Ca^2+^ homeostasis to improve insulin secretion from pancreatic β-cells in a Wolfram syndrome cellular model [87]. Although the efficacy cannot be compared with that of dantrolene as clinical trials have not yet been conducted, it supports the possibility that Ca^2+^ stabilizers could be candidate drugs for Wolfram syndrome treatment.

Chemical chaperones, such as 4-phenylbutyric acid (4-PBA) and tauroursodeoxycholic acid (TUDCA), which are known to reduce ER stress by stabilizing protein folding, have been used for treatment in Wolfram syndrome and T2DM models. We recently reported the preclinical study data on a combination of 4-PBA and TUDCA treatment, indicating that these drugs effectively stabilized WFS1 protein and improved patient-derived pancreatic β-cell functionality [85]. Based on these findings, we plan a clinical study with a combination of 4-PBA and TUDCA treatment. A chemical chaperone that enhances ER function can also normalize hyperglycemia and systemic insulin sensitivity and alleviate the increased ER stress in leptin-deficient (ob/ob) mice, a T2DM experimental mouse model [88]. Thus, chaperone-mediated restoration of proper protein folding improves β-cell function and glucose homeostasis in Wolfram syndrome and T2DM.

As noted, shared mechanisms lead to the development of Wolfram syndrome, T1DM, and T2DM. A drug commonly used to treat T2DM, glucagon-like peptide (GLP)-1 receptor agonist, also attenuates diabetes and neurodegeneration in Wolfram syndrome [89,90,91,92,93,94,95]. GLP-1 is an incretin hormone that stimulates insulin secretion after an oral glucose administration. Similar to endogenous GLP-1, the GLP-1 receptor agonist promotes insulin secretion from pancreatic β-cells and inhibits glucagon secretion from pancreatic α-cells. GLP-1 receptor agonist has been reported to attenuate ER stress in pancreatic β-cells and improve insulin secretion in Wolfram syndrome [93,94,95]. Furthermore, surprisingly, the GLP-1 receptor agonist improves cognitive, visual, and auditory function in a rat model of Wolfram syndrome [89,90,92]. GLP-1 receptor agonist is an excellent example of translating therapeutics commonly used in T2DM for the treatment of Wolfram syndrome.

Valproic acid is a therapeutic candidate for the treatment of Wolfram syndrome approached from a completely different perspective. Valproic acid is a drug used to treat bipolar disorder, and we have previously shown that valproic acid enhances the gene expression of *WFS1* [96]. In addition, Valproate was shown to enhance the dissociation of WFS1 and GPR94, a component of the UPR involved in protein folding and degradation. Based on these findings, a phase 2 randomized, double-blind clinical trial is ongoing to evaluate the efficacy of sodium valproate as a treatment for Wolfram syndrome patients’ vision loss and neurological symptoms (NCT number, NCT03717909).

Overall, these data indicate that pharmacologic manipulation of UPR may provide a novel therapeutic target not only for Wolfram syndrome but also for T2DM.

### 4.2. Regenerative Therapy

In recent years, the clinical application of cell replacement therapy in diabetes has been under development. Attempts have been made to restore insulin capacity by regenerating pancreatic β-cells in diabetic patients with impaired function. iPSCs were generated from various diabetic patients, and it became possible to differentiate iPSCs lines into stem cell-derived pancreatic β-cells (SC-β-cells). As a result, several SC-β-cells have been generated from T1DM patients, and studies for clinical application have been underway. The first phase 1/2 study of SC-β-cells transplantation in T1DM patients is ongoing (NCT number, NCT04786262).

SC-β-cells or β-like cells differentiated from patient-derived iPSCs are also available for replacement therapy in Wolfram syndrome [97,98]. We have previously shown that Wolfram syndrome-patient iPSCs have a low differentiation efficiency into SC-β-cells and reduced insulin secretion, whereas corrected SC-β-cells, in which the pathogenic variant of *WFS1* was repaired by CRISPR/Cas9, showed marked insulin secretion recovery [97]. In addition, corrected SC-β-cell transplantation recovered normal glucose homeostasis in streptozotocin-induced diabetic mice. Although there are still many challenges to overcome, such as developing transplantation methods, the current availability of genetically engineered SC-β-cells could be a therapeutic breakthrough for monogenic forms of diabetes, such as Wolfram syndrome. Multiple iPSCs have been generated from patients with monogenic diabetes to elucidate the function of genes that, when mutated, increase the risk for the development of T2DM [99]. Although there are no reports describing the generation of SC-β-cells from T2DM patient-derived iPSCs, there are several reports of differentiating iPSCs into β-like cells [100,101,102,103]. With further progress in identifying protective or risk genes for T2DM [100,101,102], we can expect increased utilization of regenerative therapy methods, including transplantation of gene-edited SC-β-cells to T2DM patients.

## 5. Conclusions and Future Perspectives

Excess ER stress in pancreatic β-cells is involved in the pathogenesis of T2DM, and dysfunction of multiple UPR-related genes, including *WFS1*, are candidate genetic risk factors for T2DM. Creating an experimental disease model is essential for studying the genetic risk factors for T2DM development; however, the heterogeneity of T2DM makes this task difficult. Since Wolfram syndrome is a monogenic form of diabetes caused by mutations in *WFS1*, Wolfram syndrome is a prototype for ER stress-related diseases such as T2DM. To understand the genetic background of T2DM, we can utilize the Wolfram syndrome experimental models that have already been established, including SC-β-cells.

In the future, we expect to develop individualized medical care for Wolfram syndrome patients, prescribing effective drugs selected based on their personal *WFS1* variant. Moreover, regenerative medicine using SC-β-cells and gene editing technology would make it possible to transplant the patient’s tissue in a “normal” condition. For T2DM, if more risk genes are identified in the future, it would also become possible to develop regenerative therapies.

## Figures and Tables

**Figure 1 ijms-24-00230-f001:**
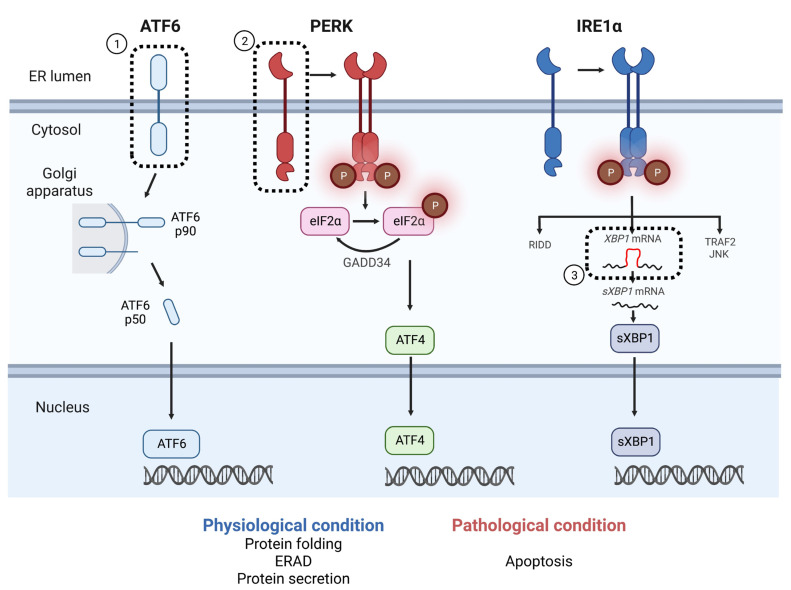
UPR mechanism and the reported SNPs associated with T2DM. The three major regulators (ATF6α, PERK, and IRE1α) are involved in the UPR pathway under pathophysiological conditions. On sensing ER stress, ATF6α (p90) is cleaved at the Golgi apparatus. Cleaved ATF6α (p50) translocates to the nucleus and acts as a transcription factor. PERK arrests protein synthesis by phosphorylating eukaryotic translation initiation factor 2 subunit α (eIF2α). Phosphorylated eIF2α also induces the expression of ATF4, which regulates protein synthesis, apoptosis, and autophagy. IRE1α forms oligomers and autophosphorylates on sensing ER stress, followed by the acquisition of RNase activity. GADD33 dephosphorylates eIF2α and restores protein synthesis. The activated IRE1α splices the *XBP1* intron and induces the expression of the transcription factor, spliced *XBP1* (*sXBP1*). IRE1α also suppresses protein folding loads by degrading mRNA, called regulated IRE1-dependent decay (RIDD). In addition, IRE1α activates the TRAF2/JNK pathway followed by the activation of the autophagy pathway. ①–③: The molecules whose SNPs have been reported to contribute to the development of T2DM or insulin resistance include the following. ① *ATF6* SNPs, detected in a population of Pima Indians and Dutch Caucasians [12,13]; ② *PERK* SNPs, detected in Han Chinese populations correlated with increased risk of prediabetes, elevated BMI, and insulin resistance [14]; ③ *XBP1* SNPs, reported at high frequency in Han Chinese populations with T2DM and impaired glucose regulation [14,15]—ATF6α (or ATF4), activating transcription factor 6α (or 4); PERK, protein kinase R-like ER kinase; inositol requiring enzyme 1α (IRE1α); ERAD, ER-associated degradation.

**Table 1 ijms-24-00230-t001:** Genetic Variants in UPR Pathways that are Related to Diabetes.

UPR Pathway	Gene	Protein	Syndrome	Symptoms Other than Diabetes
PERK	*EIF2AK3* [32,33,34]	PERK	Walcott–Rallison syndrome	Liver dysfunction Skeletal dysplasia
	*EIF2S3* [44,45]	eIF2 γ subunit	Mental retardation, epileptic seizures, hypogonadism, Hypogenitalism, microcephaly, and obesity (MEHMO) syndrome	Mental retardation Epilepsy Hypogonadism/hypogenitalism Microcephaly and obesity
	*EIF2B1* [46]	eIF2B		Liver dysfunction
	*DNAJC3* [47]	p58^IPK^	Ataxia, combined cerebellar and peripheral, with hearing loss, and diabetes mellitus (ACPHD)	Combined cerebellar Afferent ataxia Mild upper motor neuron damage Peripheral neuropathy Sensorineural hearing loss
	*PPP1R15B* [48]	CReP		Short stature Intellectual disability Microcephaly
IRE1	*ERN1* [36,37] ^#1^	IRE1α		
Other	*WFS1* [49]	Wolframin	Wolfram syndrome, Wolfram syndrome-related disorder	Optic nerve atrophy, hearing loss, diabetes insipidus, neurodegeneration
	*IER3IP1* [50,51]	IER3IP1	Microcephaly with simplified gyration, epilepsy, and permanent neonatal diabetes syndrome (MEDS)	Microcephaly Epilepsy
	*CISD2* [52]	ERIS	Wolfram syndrome (type 2)	Upper gastrointestinal ulceration and bleeding
	*MANF* [42,43]	MANF		Short stature, hearing loss, developmental delay, microcephaly
	*CREBRF* [53] ^#2^	CREB3		
	*YIPF5* [54]	YIPF5		Microcephaly Epilepsy
	*TANGO1* [55]	TANGO1		Dentinogenesis imperfecta Short stature Skeletal abnormalities
	*SIL1* [56] ^#1^	SIL1	Marinesco–Sjögren syndrome	Early-onset cerebellar ataxia Short stature
	*ERDJ4* [57] ^#1^	ERDJ4		

^#1^ Diabetic phenotype was reported only in mouse; ^#2^ Decreased the risk of T2DM in Samoans.

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
