# Peer review of "The Role of ER Stress in Diabetes: Exploring Pathological Mechanisms Using Wolfram Syndrome"

_ijms, 2022, doi:10.3390/ijms24010230_

Round 1
Reviewer 1 Report
The authors discuss dysregulated ER stress in T2DM pathogenesis including new possible treatments targeting protein folding and ER stress pathways with particular focus on pivotal studies of Wolfram syndrome. The paper flows well and is concise. I would only suggest to make reference to other human studies, apart from Kondo et al, involving Liraglutide use in Wolfram Syndrome type 1 since has not only been assessed in animal models.
Author Response
We appreciate the valuable feedback. Based on the suggestion, we added some references and descriptions. All the changes are highlighted in yellow.

Reviewer 2 Report
This is an excellent and very comprehesive and easily comprehensible article. It has showcased an important link beteween ER stress, diabetes nad explored the causals via the wolfram syndrome.
The flow of the article helps the reader easily follow through and the extensive references will aslo help further understanding. This is going to be highly cited and is an important service for students and advanced researchers in the field.
Author Response
We appreciate these comments from the reviewer.
